

# Nitrous oxide, methane emissions and grain yield in rainfed wheat grown under nitrogen enriched biochar and straw in a semiarid environment

Stephen Yeboah[1,2,3,*], Wu Jun[2,3,*], Cai Liqun[2,3], Patricia Oteng-Darko[1], Erasmus Narteh Tetteh[1,4] and Zhang Renzhi[2,3]

[1] CSIR-Crops Research Institute, Kumasi, Ghana
[2] College of Resources and Environmental Sciences, Gansu Agricultural University, Gansu, China
[3] Gansu Provincial Key Lab of Arid Land Crop Science, Gansu Agricultural University, Gansu, China
[4] Kwame Nkrumah University of Science and Technology (KNUST), Kumasi, Ghana
[*] These authors contributed equally to this work.

Corresponding authors
Stephen Yeboah,
proyeboah@yahoo.co.uk
Zhang Renzhi, zhanrenzi@gmail.com

## ABSTRACT

**Background.** Soil application of biochar and straw alone or their combinations with nitrogen (N) fertilizer are becoming increasingly common, but little is known about their agronomic and environmental performance in semiarid environments. This study was conducted to investigate the effect(s) of these amendments on soil properties, nitrous oxide ($N_2O$) and methane ($CH_4$) emissions and grain and biomass yield of spring wheat (*Triticum aestivum* L.), and to produce background dataset that may be used to inform nutrient management guidelines for semiarid environments.

**Methods.** The experiment involved the application of biochar, straw or urea (46% nitrogen [N]) alone or their combinations. The treatments were: $CN_0$ –control (zero-amendment), $CN_{50}$ –50 kg ha$^{-1}$ N, $CN_{100}$ –100 kg ha$^{-1}$ N, $BN_0$ –15 t ha$^{-1}$ biochar, $BN_{50}$ –15 t ha$^{-1}$ biochar + 50 kg ha$^{-1}$ N, $BN_{100}$ –15 t ha$^{-1}$ biochar + 100 kg ha$^{-1}$ N, $SN_0$ –4.5 t ha$^{-1}$ straw, $SN_{50}$ –4.5 t ha$^{-1}$ straw + 50 kg ha$^{-1}$ N and $SN_{100}$ –4.5 t ha$^{-1}$ straw + 100 kg ha$^{-1}$ N. Fluxes of $N_2O$, $CH_4$ and grain yield were monitored over three consecutive cropping seasons between 2014 and 2016 using the static chamber-gas chromatography method.

**Results.** On average, $BN_{100}$ reported the highest grain yield (2054 kg ha$^{-1}$), which was between 25.04% and 38.34% higher than all other treatments. In addition, biomass yield was much higher under biochar treated plots relative to the other treatments. These findings are supported by the increased in soil organic C by 17.14% and 21.65% in biochar amended soils (at 0–10 cm) compared to straw treated soils and soils without carbon respectively. The $BN_{100}$ treatment also improved bulk density and hydraulic properties ($P < 0.05$), which supported the above results. The greatest $N_2O$ emissions and $CH_4$ sink were recorded under the highest rate of N fertilization (100 kg N ha$^{-1}$). Cumulative $N_2O$ emissions were 39.02% and 48.23% lower in $BN_{100}$ compared with $CN_0$ and $CN_{100}$, respectively. There was also a ≈ 37.53% reduction in $CH_4$ uptake under $BN_{100}$ compared with $CN_0$–control and $CN_{50}$. The mean cumulative $N_2O$ emission from biochar treated soils had a significant decrease of 10.93% and 38.61% compared to straw treated soils and soils without carbon treatment, respectively. However, differences between mean cumulative $N_2O$ emission between straw treated soils and

soils without carbon were not significant. These results indicate the dependency of crop yield, $N_2O$ and $CH_4$ emissions on soil quality and imply that crop productivity could be increased without compromising on environmental quality when biochar is applied in combination with N-fertilizer. The practice of applying biochar with N fertilizer at $100 \, \text{kg ha}^{-1}$ N resulted in increases in crop productivity and reduced $N_2O$ and $CH_4$ soil emissions under dryland cropping systems.

# INTRODUCTION

Atmospheric methane ($CH_4$) and nitrous oxide ($N_2O$) are persistent greenhouse gases (GHG) influencing global warming (*IPCC, 2014*). Agriculture contributes significant amounts of $N_2O$ and $CH_4$ to the atmosphere, however net GHG emissions as $CO_2$ from farming-related activities can be potentially reduced by increasing carbon (C) sequestration in soil and crop biomass (*Wang et al., 2021*). This may be achieved by implementing improved crop and fertilizer management practices that maximize biomass production and C returned to soil (*Norton, 2014*). There are no significant terrestrial sinks of $N_2O$ hence reduction in its emission may only be achieved by managing nitrogen (N) inputs, and improving soil conditions and efficiency of applied fertilizer-N (*Grace, 2016*). However, in semi-arid regions of China, in an attempt to increase yields, farmers are compelled to apply more fertilizer, leading to an over-application (*Xu & Yang, 2017*). There is heavy dependence on mineral fertilizers to ensure adequate N supply for crops, and in most cases more fertilizer is applied than needed by the plant (*Liu et al., 2016*). This is a common practice in most farming communities in semi-arid regions of China (*Wang et al., 2021*). The situation has led to negative impact on the environment, and threatens the long-term sustainability of Chinese agriculture (*Liu et al., 2016*; *Wang et al., 2021*). Therefore, it is key to identify suitable agricultural practices that could help maximize crop production without compromising on environmental quality.

Current increases in atmospheric GHG levels require that novel approaches are undertaken to mitigate impacts of climate change, such as management practices capable of improving soil C sequestration (*Woolf et al., 2010*). Soil carbon sequestration through application of recalcitrant C-rich biochar is mentioned as a suitable means to mitigate climate change, and improve soil fertility (*Laird et al., 2010*) and crop productivity (*Steiner et al., 2007*). According to *Saggar (2010)* $N_2O$ emissions are driven by the applications of fertilizer nitrogen (N), soil tillage and crop type, with their effects dependent on soil and weather conditions. Biochar application as a soil amendment, could therefore be an effective strategy for mitigating emissions and increasing crop yield. However, the effect of biochar on soil properties, GHG emissions and crop yield have been diverse. Several mechanisms have also been proposed in literature to explain the diverse effects, with limited amounts of evidence to support them. *Yanai, Toyota & Okazaki (2007)* reported

decreased $N_2O$ and $CH_4$ soil emissions in response to biochar application. In contrast, *Clough et al. (2013)* observed no suppression of $N_2O$ and $CH_4$ soil emissions, whilst similar effect was observed by *Zhang et al. (2010)*. *Zhang et al. (2011)* also reported that biochar application in dryland significantly reduces soil $CH_4$ emission by 33% compared to soil without biochar. *Zimmerman, Gao & Ahn (2011)* attributed the positive effect of biochar application on soil $CH_4$ emissions to the inhibition of soil methanotrophs while *Zhu et al. (2018)* associated reduced soil $CH_4$ emissions to the change in the ratio of methanogenic to methanotrophic archaea. In general, most studies have found biochar amendments to either decrease or not significantly affect soil $N_2O$ emissions; however, some few reports have found increased $N_2O$ emissions following biochar amendments (*Yeboah et al., 2018*). Explanations for continued long-term suppression of $N_2O$ emissions in biochar-amended soils include alterations in microbial communities due to physical habitat changes, physical and/or chemical protection of organic C and/or N by biochar and alteration of micro-scale soil redox status due to electrochemical properties of biochars (*Rivka, David & Timothy, 2019*). It is thus clear that, these effects have been shown to vary significantly depending upon the type of biochar used and the environmental and soil conditions under which the material is applied.

The Loess Plateau is an important agricultural area in China and is widely used for grain production (*He et al., 2014*). The area is one of the most severely eroded regions in China, which coupled with limited precipitation and high evaporation rates, often results in poor crop productivity (*He et al., 2014*). Many studies have indicated that human activities, such as land use is responsible for the degradation and loss of soil fertility in semi-arid regions of China (*Xu & Yang, 2017*; *Zhang et al., 2017*; *Huang et al., 2019*). Traditional methods of soil cultivation often accelerates the decline of soil fertility, and loss of soil organic C (*Lamptey, Li & Xie, 2018*). Given the fact that the population of semi-arid regions in China mainly relies on rainfed agriculture for their livelihood; developing environmentally friendly and sustainable nutrient management strategies is crucial. There is limited information on the specific impact of widely-used agronomic practices involving biochar, straw and nitrogen fertilizer used alone or combined on greenhouse gas emission and crop yield in drier lossiah soils (*Solomon et al., 2007*). Moreover, little is known about the effect of biochar application to soil under arid conditions (*Arfaoui, Ibrahimi & Trabelsi, 2019*). This study hypothesized that increased C inputs would raise the soils potential to reduce $N_2O$ and $CH_4$ soil emissions whilst increasing grain yield. Therefore, the objectives of this study were to: 1) determine the effect of biochar, straw and nitrogen fertilizer applied alone or combined with fertilizer-N on soil properties, (2) assess the effect of biochar, straw and nitrogen fertilizer applied alone or combined with fertilizer-N on biomass and grain yield of spring wheat, and (3) determine the effects of biochar, straw and nitrogen fertilizer used alone or combined with fertilizer-N on $N_2O$ and $CH_4$ emissions.

## MATERIALS & METHODS

### Study site

The study was conducted during the 2014, 2015 and 2016 growing seasons at the Dingxi Experimental Station (35°28′N, 104°44′E, elevation 1971-m above-sea-level) of the Gansu

Agricultural University in Northwestern China. The research station is located in the semiarid Western Loess Plateau, which is characterized by step hills and deeply eroded gullies (*Feng et al., 2013*). This area has Aeolian soils, locally known as Huangmian (*Chinese Soil Taxonomy Cooperative Research Group, 1995*), which equate to Calcaric Cambisols based on the *FAO (1990)* description. The soil type in the study area is sandy-loam with low fertility. The soil has a pH of $\approx$8.3, soil organic carbon (SOC) $\leq$8.13 g kg$^{-1}$, and Olsen-P $\leq$13 mg kg$^{-1}$ as described in *Yeboah et al. (2018)*. The type of soil in the study area is the principal soil for cultivation of crops in the agro-ecological zone. Long term average rainfall, evaporation and aridity in the study area is 391.9 mm per annum; 1531 mm per annum and 2.53 respectively. The aridity index (AI) is the degree of dryness of the climate at the study area. In July, the daily maximum temperature can increase to 38 °C. Similarly, in January daily minimum temperature can drop to $-22$ °C. Annual cumulative temperatures >10 °C are 2240 °C and annual radiation is 5930 MJ m$^{-2}$, with 2477 h of sunshine as described in *Yeboah et al. (2018)*. The agro-climatic conditions are similar to semiarid environments. The research site is characterized by continuous cultivation of the same field using conventional tillage practices. The preceding crop cultivated at the research site was potatoes (*Solanum tuberosum* L.). Seasonal rainfall recorded in 2014, 2015 and 2016 during the research was 174.6, 252.5 and 239.4 mm respectively (Fig. 1).

**Experimental design and description of treatment**

The experiment involved addition of different carbon (C) sources; namely: biochar and straw, and N fertilizer in the form of urea (46% N) arranged in a randomized block design with nine treatments and three replications (*Yeboah et al., 2018*). The treatments were: $CN_0$–control (zero-amendment), $CN_{50}$ –50 kg ha$^{-1}$ N applied each year, $CN_{100}$ –100 kg ha$^{-1}$ N applied each year, $BN_0$ –15 t ha$^{-1}$ biochar applied in a single dressing in 2014, $BN_{50}$ –15 t ha$^{-1}$ biochar applied in a single dressing in 2014 + 50 kg ha$^{-1}$ N applied each year, $BN_{100}$ –15 t ha$^{-1}$ biochar applied in single dressing in 2014 + 100 kg ha$^{-1}$ N applied each year, $SN_0$ –4.5 t ha$^{-1}$ straw applied each year, $SN_{50}$ –4.5 t ha$^{-1}$ straw applied each year + 50 kg ha$^{-1}$ N applied each year and $SN_{100}$ –4.5 t ha$^{-1}$ straw applied each year + 100 kg ha$^{-1}$ N applied each year as described in *Yeboah et al. (2018)*. The two C sources (biochar and straw) were applied at the same quantity based on the straw returned to the soil every year and straw C mineralization. Biochar was spread evenly on the soil surface in March 2014 and incorporated into the soil using a rotary tillage implement to a depth of $\approx$10 cm. The biochar was obtained from Golden Future Agriculture Technology Company Limited, Liaoning in China. Biochar was produced from maize straw using pyrolysis process at a temperature of 350–550 °C. This process converted about 35% of the maize straw to biochar. The biochar in the form of granules was milled to a size of <5 mm to allow for even mixing with the soil. The wheat crop of the previous season from the research station was used as a source of straw for the study. In the plots that received straw treatment, the straw from the previous wheat crop was weighed and returned to the original plots. This was done after threshing. Biochar analysis was conducted using the procedure as describe in *Lu (2000)*. Total C and N and soil pH were determined using a CN Analyzer (analytikjena; multi N/C, 2100S, Germany) and Kjeldahl digestion and distillation (*Bremner & Mulvaney,*

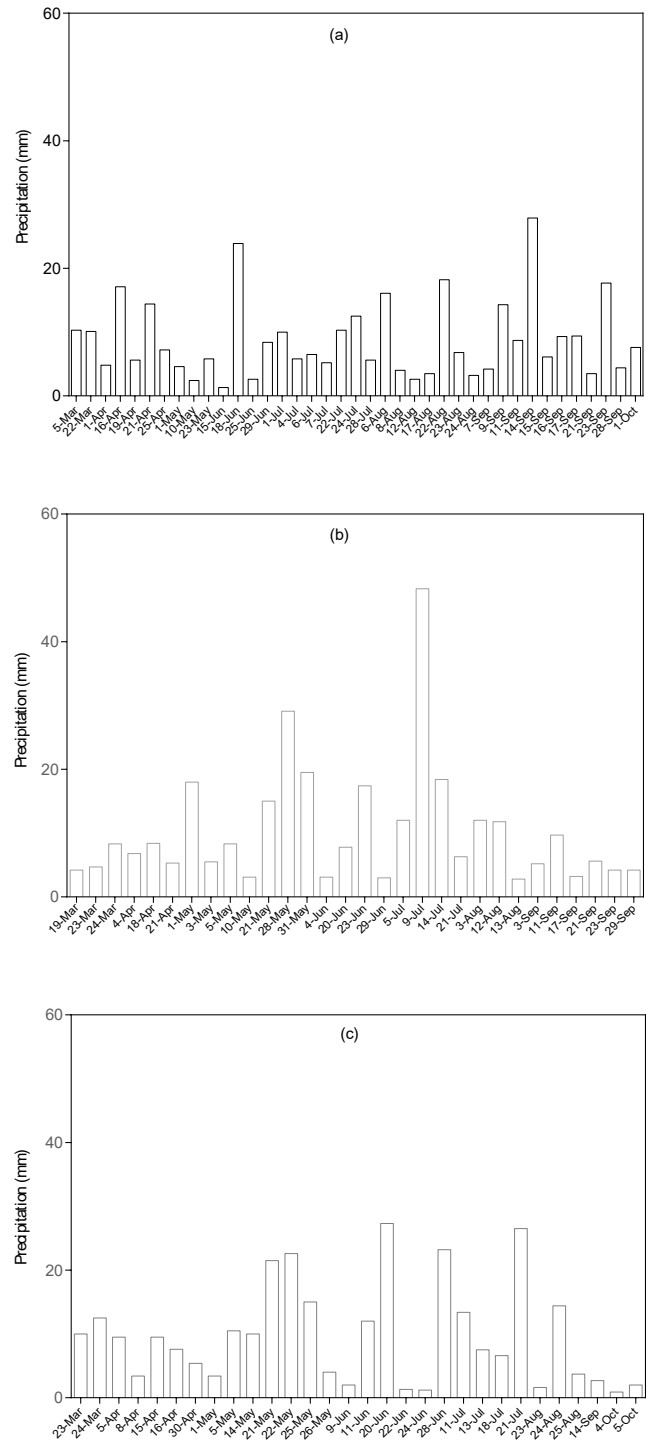

**Figure 1** Precipitation (mm) in (A) 2014, (B) 2015 and (C) 2016 cropping season at the experimental site.

**Table 1  Characterization of biochar and straw used in the study.**

| Parameter | pH | BD (g cm⁻³) | SA (m² g⁻¹) | Ca | Mg | K | C | N | P | Ash content (%) |
|---|---|---|---|---|---|---|---|---|---|---|
| | | | | | | | **%** | | | |
| Biochar | 9.2 | 0.68 | 8.75 | 0.8 | 0.47 | 0.51 | 53.28 | 1.04 | 0.26 | 25.5 |
| Straw | 6.5 | / | / | 0.53 | 0.04 | 0.47 | 45.05 | 0.94 | 0.08 | 8.9 |

**Notes.**
Values are means for $n = 2$.

*1982*) and pH meter (model: Sartorius PB–10, Germany). The soil pH was determined using soil to water ratio of 1: 2.5. Similar protocol was used to determined total C and N, ash content and pH of the straw. Table 1 shows the chemical characterization of biochar and straw used in the experiment. All the treatments received a blanket application of Phosphorus (P) fertilizer which was applied equally at a rate of 46 kg ha⁻¹ P in the form of ammonium dihydrogen phosphate (12% N, 52% $P_2O_5$). No–tillage seeder was used to incorporate the fertilizer to about 20 cm soil depth at planting. Based on the protocol described in *Yeboah et al. (2016)* Spring wheat (*Triticum aestivum* L. cv. Dingxi 35) was sown in Mid-March at a rate of 188 kg ha⁻¹ seeds at 20-cm row spacing. The crop was harvested either at the end of July or early August. The individual plot's measured 3 m by 6 m and the plots were separated by 0.5 m width protection rows.

## Soil sampling, measurements and analyses

Based on the protocol described in *Yeboah et al. (2016)*, soil bulk density (BD) was determined by taking small cores and relating the oven–dried mass of soil to the volume of the core. Soil saturated hydraulic conductivity (Ksat) was determined at two points per plot using the disc permeameter method according to *Carter (1993)*. Soil samples were collected from 0–10 and 10–30 cm depth and bulked for analysis. The samples were processed for analysis using the protocol described in *Yeboah et al. (2018)*. Soil organic carbon (SOC) in the fine ground samples was determined by the modified *Walkley & Black (1934)* wet oxidation method (*Nelson & Sommers, 1982*).

## Gas sampling and analysis

Collection of $N_2O$ and $CH_4$ gases were performed using the static chamber technique based on the procedure described by *Zou et al. (2005)*. For each sampling event, gas collection was consistently performed between 08:00–12:00 h, based on the guidelines of *Yeboah et al. (2016)*. Collection of samples for $N_2O$ and $CH_4$ analyses was conducted at 0, 10, and 20 min after chamber closure. Samples were collected between March and September and detailed sampling procedure could be found in *Yeboah et al. (2016)*; *Yeboah et al. (2018)*. Based on earlier studies conducted in low rainfall areas (*e.g.*, *Wang et al., 2010*) emissions occurring during the dry season were expected to be low and therefore did not justify measurements over that period. Gas fluxes were measured over 14 sampling events per year. Whilst acknowledging that accurate estimates of total emissions cannot be determined from relatively few sampling events, the main purpose of this work was to quantify relative differences between-treatments, which therefore justifies the approach used in this study. A similar approach was also employed by *Tullberg et al. (2018)* to quantify soil emissions

of GHG from tillage and traffic treatments in conservation agriculture areas with seasonal rainfall. The $N_2O$ and $CH_4$ concentration in samples were analyzed within 2 to 3 days after collection using gas chromatograph (GC). The GC system (Agilent 7890A, USA) equipped with flame ionization detector (FID) was used for $CH_4$ analysis and an electron capture detector (ECD) was used for $N_2O$ analysis. Rates of $CH_4$ and $N_2O$ fluxes were calculated by linear increment of the gas concentration at 0, 10 and 20 min. The calculation was only accepted when the $R^2$ of the linear correlation was higher than 0.90 ($p < 0.05$). The average GHG fluxes were a mean of three replicates of each treatment over the sampling dates. Further procedure for the analysis and conditions of the column could be found in *Yeboah et al. (2018)* and *Zou et al. (2005)*.

### Estimations of nitrous oxide and methane emissions

The $N_2O$ (mg m$^{-2}$ h$^{-1}$) and $CH_4$ (mg m$^{-2}$ h$^{-1}$) emissions were calculated using Eq. (1) based on the protocol described in *Yeboah et al. (2016)*:

$$F = \frac{C_2 \times V \times M_0 \times 273/T_2 - C_1 \times V \times M_0 \times 273/T_1}{A \times (t_2 - t_1) \times 22.4} \tag{1}$$

where: F are fluxes of $N_2O$ or $CH_4$ (mg m$^{-2}$ h$^{-1}$), V is volume (m$^3$), $M_0$ is the molecular weight of the gas, $C_1$ and $C_2$ are the concentration of previous (0 mins) and current (20 mins) gas concentrations inside the chamber (mol mol$^{-1}$), $T_1$ and $T_2$ are temperature (Kelvin) recorded inside the chamber during current and previous samplings, and $t_1$ and $t_2$ are previous and current sampling times (h).

The cumulative emission of $N_2O$ and $CH_4$ in kg ha$^{-1}$ was estimated using the equation as follows (*Yeboah et al., 2016*):

$$M = \sum (F_{N+1} + F_N) \times 0.5 \times (t_{N+1} - t_N) \times 24 \times 10^{-2} \tag{2}$$

where M is the $N_2O$ and $CH_4$ cumulative emissions during the period of measurement (kg ha$^{-1}$), F is $N_2O$ and $CH_4$ emission (in mg m$^{-2}$ h$^{-1}$); and previous and current sampling emissions were N+1 and N respectively. The number of days from first sampling is represented by t.

### Biomass and grain yield

Biomass and grain yield was determined by cutting the plants using hand sickles to five cm height aboveground. The outer edges of about 0.5 m was discarded from each plot. Both yields were determined on a dry–weight basis by oven–drying the plant material at 105 °C for 45 min and then to constant weight at 85 °C (*Yeboah et al., 2016*).

### Statistical analyses

Statistical analyses were undertaken with the SPSS 22 (IBM Corporation, Chicago, IL, USA) with the treatment as the fixed effect and year as random effect. Tukey's honestly significant was used to determine the differences between-treatments means. Significance differences were declared at probability level of 5%.

**Table 2  Analysis of variance for carbon, nitrogen and year effects and their interaction.**

| Sources | Soil bulk density | | | Soil organic carbon | | | N$_2$O | CH$_4$ | Biomass yield | Grain yield |
|---|---|---|---|---|---|---|---|---|---|---|
| | 0–5 | 5–10 | Ksat | 0–5 | 5–10 | 10–30 | | | | |
| Carbon (C) | ** | * | * | * | * | n.s. | ** | n.s. | ** | ** |
| Nitrogen (N) | ** | n.s. | * | ** | * | n.s. | ** | n.s. | ** | ** |
| Year (Y) | n.s. | * | * | * | n.s. | n.s. | * | ** | * | ** |
| C ×N | n.s. | n.s. | n.s. | n.s. | n.s. | n.s. | n.s. | * | * | n.s |
| C ×Y | n.s. | n.s. | n.s. | ** | ** | n.s. | n.s. | ** | n.s | n.s |
| N ×Y | n.s. | n.s. | n.s. | n.s. | n.s. | n.s. | n.s. | n.s. | n.s | ** |

Notes.
*, ** Indicate significant difference at $P < 0.05$ and $P < 0.01$, respectively. n.s. indicate no significance difference at $P < 0.05$.

# RESULTS

## Soil bulk density, saturated hydraulic conductivity and soil organic carbon

Soil samples taken during the study period showed significant differences in the bulk density depending on the type of treatment and the depth of sampling (Table 2). Bulk density increased with soil depth in many cases irrespective of treatment over the experimental period. Significant differences between treatments were minor in the upper layer in 2014, but significant treatment effect was recorded in the 5–10 cm soil depth as the experimental period progressed from 2014 to 2016 (Table 3). On average, the lowest bulk density (1.14 g cm$^{-3}$) was recorded under biochar–amended soils, and the highest was observed under soils with carbon (1.21 g cm$^{-3}$). The results obtained with the straw–amended soils showed a similar trend, except that differences were not significant at $p < 0.05$ in most cases. Saturated hydraulic conductivity (Ksat) was significantly ($p < 0.05$) affected by carbon, N fertilizer and year but there was no significant interaction between treatment factors (Table 2). Application of BN$_{100}$ treatment enhanced mean saturated hydraulic conductivity by 23.7%, 24.3% and 20.4% relative to CN$_0$, CN$_{50}$ and SN$_0$, respectively (Table 4). Carbon and year had significant interaction ($p < 0.05$) on soil organic carbon, except at the 10–30 cm soil depth (Table 2). Similarly, carbon and fertilizer-N also interactively affected soil organic C in all the soil depth evaluated. Application of fertilizer-N at the 50 and 100 kg ha$^{-1}$ rate influenced SOC significantly ($p < 0.05$) under biochar treated soils, particularly in the depth of 0–5 cm (Table 5). However, N$_{100}$ had greater effect compared to N$_{50}$.

## Nitrous oxide emissions

All the treatments were sources of nitrous oxide (N$_2$O) emission throughout the sampling period and the maximum observed N$_2$O emissions occurred in early July in each year of this study (Fig. 2). These responses were consistent with recorded soil moisture and temperature data. Significant differences ($p < 0.05$) were found among treatments at certain periods of measurement (Fig. 3). For example, in 2014, the maximal N$_2$O emission of BN$_{100}$ was 79.5 µg m$^{-2}$ h$^{-1}$ and the minimal was 36.5 µg m$^{-2}$ h$^{-1}$; they were significantly lower than those for CN$_{50}$ (100.7 µg m$^{-2}$ h$^{-1}$ for maximum and 55.8 µg m$^{-2}$ h$^{-1}$ for minimum) and CN$_0$ (98.1 µg m$^{-2}$ h$^{-1}$ for maximum and 50.2 µg m$^{-2}$ h$^{-1}$ for minimum). At a lesser

**Table 3  Soil bulk density as affected by carbon addition sources.**

| Treatment | | Soil BD (g cm$^{-3}$) | | | | | |
| C source | Mineral N | 0–5 | 5–10 | | | | 10–30 |
| | | | | cm | | | |
| | | Mean | 2014 | 2015 | 2016 | Mean | Mean |
| No carbon | $N_0$ | 1.24a | 1.32a | 1.30ab | 1.27a | 1.29a | 1.29a |
| | $N_{50}$ | 1.17bc | 1.24a | 1.25abc | 1.17bc | 1.22abc | 1.24a |
| | $N_{100}$ | 1.20ab | 1.20a | 1.17cd | 1.12c | 1.16bc | 1.27a |
| Biochar | $N_0$ | 1.17bc | 1.25a | 1.25abc | 1.24ab | 1.25abc | 1.27a |
| | $N_{50}$ | 1.13cd | 1.24a | 1.16cd | 1.14c | 1.18bc | 1.24a |
| | $N_{100}$ | 1.15bcd | 1.21a | 1.18bcd | 1.17bc | 1.19bc | 1.27a |
| Straw | $N_0$ | 1.21ab | 1.22a | 1.32a | 1.23ab | 1.25ab | 1.29a |
| | $N_{50}$ | 1.11d | 1.24a | 1.08d | 1.14c | 1.16c | 1.21a |
| | $N_{100}$ | 1.14cd | 1.28a | 1.18cd | 1.16bc | 1.21abc | 1.25a |

**Notes.**
Values with different letters within a column are significantly different at $P < 0.05$.

**Table 4  Saturated hydraulic conductivity as affected by carbon addition sources.**

| Treatment Saturated hydraulic conductivity (mm h$^{-1}$) | | | | | |
| C source | Mineral N | 2014 | 2015 | 2016 | mean |
| No carbon | $N_0$ | 62.64b | 68.86b | 62.95c | 64.82c |
| | $N_{50}$ | 65.77ab | 64.06b | 63.71c | 64.51c |
| | $N_{100}$ | 67.63ab | 60.58b | 75.45abc | 67.89ab |
| Biochar | $N_0$ | 71.93ab | 62.88b | 68.07bc | 67.63ab |
| | $N_{50}$ | 80.49a | 70.94ab | 78.98ab | 76.80ab |
| | $N_{100}$ | 78.78ab | 78.99a | 82.79a | 80.19a |
| Straw | $N_0$ | 68.66ab | 65.65b | 65.42c | 66.58bc |
| | $N_{50}$ | 76.07ab | 66.02b | 74.75abc | 72.28ab |
| | $N_{100}$ | 75.65ab | 72.44ab | 71.24abc | 73.11ab |

**Notes.**
Values with different letters within a column are significantly different at $P < 0.05$. $n = 3$.

magnitude, $SN_0$ and $SN_{50}$ also produced significantly lower $N_2O$ emission compared to $CN_0$ and $CN_{50}$. During this period the lowest seasonal $N_2O$ emission was mostly recorded in the biochar treated soils and at a lesser magnitude in the straw treated soils.

There were no significant treatment interactions ($p < 0.05$) effect on cumulative $N_2O$ emission (Table 6), but treatment factors independently influenced cumulative $N_2O$ emission. The highest cumulative $N_2O$ emissions were consistently observed in the fertilized soils compared to the unfertilized soils, but differences were not always significant (Table 6). Application of $BN_0$, $BN_{50}$ and $BN_{100}$ significantly decreased cumulative $N_2O$ emission by 48.42%, 37.12% and 35.80% on average compared to $CN_{100}$, respectively (Table 4). The mean cumulative $N_2O$ emission of biochar was averaged at 1.83 kg ha$^{-1}$ representing significant decrease of 10.93% and 38.61% compared to straw treated soils (2.03 kg ha$^{-1}$)

**Table 5  Soil organic carbon as affected by different treatments.**

| Treatment | | Soil organic C (g kg$^{-1}$) | | | | | | | |
|---|---|---|---|---|---|---|---|---|---|
| C source | N rate | 0–10 | | | | 10–30 | | | |
| | | 2014 | 2015 | 2016 | Mean | 2014 | 2015 | 2016 | Mean |
| No carbon | $N_0$ | 9.64c | 9.86c | 10.43e | 9.98d | 9.29b | 9.58b | 9.48e | 9.45c |
| | $N_{50}$ | 10.18bc | 9.92bc | 11.54d | 10.55cd | 10.34ab | 9.73b | 10.71d | 10.26bc |
| | $N_{100}$ | 10.32bc | 10.90bc | 11.70d | 10.97bcd | 9.76ab | 10.10b | 11.05cd | 10.30bc |
| Biochar | $N_0$ | 11.82ab | 10.28bc | 14.91b | 12.34b | 10.45ab | 10.27b | 12.47b | 11.06b |
| | $N_{50}$ | 12.21ab | 14.04a | 16.01a | 14.09a | 11.59a | 12.66a | 14.41a | 12.89a |
| | $N_{100}$ | 12.42a | 14.09a | 16.26a | 14.26a | 11.04ab | 13.75a | 15.41a | 13.40a |
| Straw | $N_0$ | 9.71c | 10.14bc | 11.41d | 10.42cd | 9.58ab | 10.12b | 10.69d | 10.13bc |
| | $N_{50}$ | 10.70abc | 10.64bc | 13.77c | 11.70bc | 10.70ab | 10.41b | 11.54bcd | 10.88b |
| | $N_{100}$ | 11.08abc | 11.19b | 14.19bc | 12.15b | 10.92ab | 10.99b | 12.10bc | 11.34b |

**Notes.**

Values with different letters within a column are significantly different at $P < 0.05$. $n = 3$.

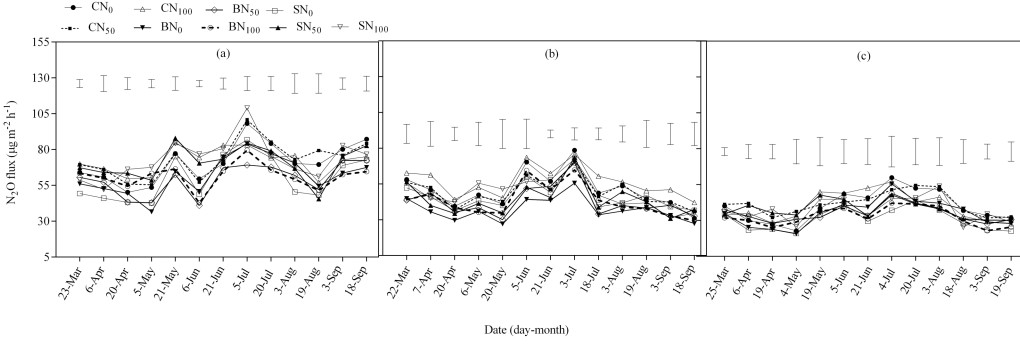

**Figure 2  Seasonal N$_2$O fluxes for spring wheat in 2014 (A), 2015 (B) and 2016 (C) as affected by carbon addition sources.**  The vertical bars represent the least significant difference (LSD) at $P < 0.05$ among treatments within a measurement date.

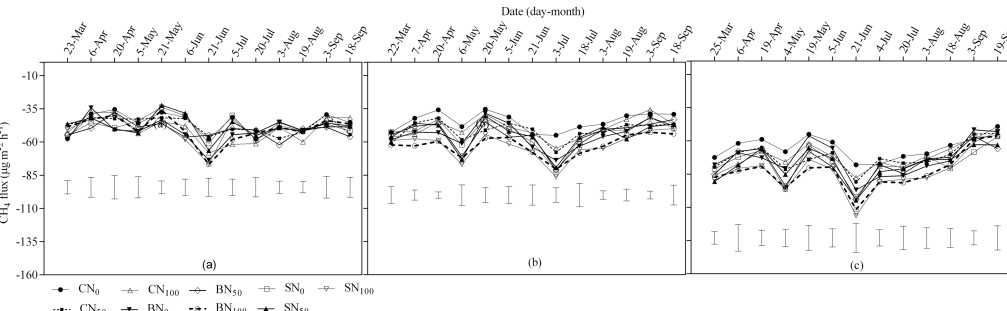

**Figure 3  Seasonal CH$_4$ fluxes for spring wheat in 2014 (A), 2015 (B) and 2016 (C) as affected by carbon addition sources.**  The vertical bars represent the least significant difference (LSD) at $P < 0.05$ among treatments within a measurement date.

**Table 6  Cumulative $N_2O$ emissions of spring wheat as affected by carbon addition sources.**

| Treatment | | $N_2O$ (kg ha$^{-1}$) | | | |
|---|---|---|---|---|---|
| C source | N rates | 2014 | 2015 | 2016 | Mean |
| No carbon | $N_0$ | 3.10a | 2.09ab | 2.00ab | 2.40ab |
| | $N_{50}$ | 3.17a | 2.00ab | 1.75bc | 2.31abc |
| | $N_{100}$ | 3.21a | 2.37a | 2.11a | 2.56a |
| Biochar | $N_0$ | 2.37b | 1.50c | 1.32c | 1.73c |
| | $N_{50}$ | 2.47b | 1.73bc | 1.41c | 1.87bc |
| | $N_{100}$ | 2.48b | 1.77bc | 1.42bc | 1.89bc |
| Straw | $N_0$ | 2.43b | 1.87bc | 1.24c | 1.85bc |
| | $N_{50}$ | 2.83ab | 1.78bc | 1.54abc | 2.05abc |
| | $N_{100}$ | 2.99a | 1.98ab | 1.61abc | 2.19abc |

**Notes.**
Values with different letters within a column are significantly different at $p < 0.05$.

and soils without carbon treatment (2.42 kg ha$^{-1}$). Straw treated soils had non–significant cumulative $N_2O$ decrease of 0.39 kg ha$^{-1}$, or 19.40% less compared to no carbon soils.

## Methane emissions

All the treatments had similar trends of seasonal $CH_4$ dynamics and were net carbon sinks over the three study years (Fig. 3). The minimum $CH_4$ consumption was recorded in April 2014 and 2015, and in September 2016. In the present study, a single peak was observed in June 2014, whiles double peaks were observed in May and July 2015 and 2016. During this period, the greatest seasonal $CH_4$ consumption of $-79.94$, $-81.07$ and $-111.59$ $\mu g\,m^{-2}\,h^{-1}$ in 2014, 2015 and 2016 respectively were observed in $BN_{100}$ soils; it was 38.14%, 47.37%, 43.05% more compared to $CN_0$ ($-57.87$, $-55.01$ and $-78.01$ $\mu g\,m^{-2}\,h^{-1}$). At a lesser extent, the maximum seasonal $CH_4$ consumption in $SN_{50}$ and $SN_{100}$ soils were significantly higher ($p < 0.05$) compared to the $CN_0$ and $CN_{50}$ soils. The results were clear that, the greater seasonal $CH_4$ consumption occurred with the higher N fertilizer soils and the greatest $CH_4$ uptake generally occurred in the biochar treated soils, followed by the straw treated soils and the least were observed in the no carbon soils

Year individually had a significant effect ($p < 0.05$) on cumulative $CH_4$ emission (Table 7), and interaction between carbon and year significantly affected cumulative $CH_4$ emission. The results of cumulative $CH_4$ emission showed that increasing N fertilizer rates generally enhanced $CH_4$ consumption in all treatments. The use of $BN_{100}$ boosted cumulative $CH_4$ uptake in 2014 (by 21.9% and 18.2%), 2015 (by 83.6% and 59.1%) and 2016 (by 30.5% and 18.4%) compared to $CN_0$ and $CN_{50}$, respectively. Increasing the fertilizer rate from $N_{50}$ to $N_{100}$ resulted in significantly higher cumulative $CH_4$ consumption ($p < 0.05$) on straw treated soils in 2014 relative to $N_0$ on soils without carbon; the increase was 16.8%. In 2015, application of $SN_{100}$ increased cumulative $CH_4$ sink by 41.0%, 73.0%, 22.8% and 26.8% compared with $CN_0$, $CN_{50}$ and $CN_{100}$, respectively. The mean cumulative $CH_4$ consumption was greatest in biochar treated plots ($-2.8$ kg ha$^{-1}$), followed by straw treated soils ($-2.6$ kg ha$^{-1}$) and the least in no carbon soils ($-2.3$ kg ha$^{-1}$).

**Table 7 Cumulative CH₄ emissions of spring wheat as affected by different treatment.**

| Treatment | | CH$_4$ (kg ha$^{-1}$) | | | |
|---|---|---|---|---|---|
| C source | N rates | 2014 | 2015 | 2016 | Mean |
| No carbon | $N_0$ | −1.80a | −1.79a | −2.83a | −2.14a |
| | $N_{50}$ | −1.85ab | −2.07a | −3.12ab | −2.35ab |
| | $N_{100}$ | −2.08bc | −2.00a | −3.13abc | −2.40abc |
| Biochar | $N_0$ | −2.09bc | −3.14c | −3.31abc | −2.85bc |
| | $N_{50}$ | −2.13bc | −2.23ab | −3.38abc | −2.58abc |
| | $N_{100}$ | −2.19c | −3.29c | −3.70c | −3.06c |
| Straw | $N_0$ | −1.91abc | −2.16ab | −3.19abc | −2.42abc |
| | $N_{50}$ | −1.96abc | −2.21ab | −3.32abc | −2.50abc |
| | $N_{100}$ | −2.10bc | −2.54b | −3.61bc | −2.75abc |

**Notes.**
Values with different letters within a column are significantly different at $P < 0.05$.

## Biomass and grain yield

There was significant interaction effects between carbon and nitrogen, and nitrogen and year on biomass yield at $p < 0.05$ (Table 2). In addition, carbon, nitrogen and year individually had significant effect on biomass yield. Application of $N_{100}$ treatments on biochar treated soils ($BN_{100}$) increased biomass yield by 39.05% in 2014, 37.31% in 2015 and 30.02% in 2016 on average compared to soils without carbon (Table 8). Similarly, $BN_{100}$ significantly increased biomass yield in 2014 (by 35.06% and 26.43%), 2015 (by 40.04% and 23.11%) and 2016 (by 21.86% and 13.45%) compared to $SN_0$ and $SN_{50}$ sites, respectively. Application of $SN_{100}$ also caused significant increases in biomass yield compared to no carbon soils, an average increase of 32.09%, 29.32% and 32.56% were recorded in 2014, 2015 and 2016 respectively. The grain yield under $N_{100}$ fertilization was significantly increased ($p < 0.05$) by 35.87%, 29.45% and 13.34% under no carbon soils; 33.64%, 37.02% and 39.16% under biochar soils, and 31.89%, 32.35% and 24.08% under biomass treated soils in 2014, 2015 and 2016, respectively, compared to their corresponding $N_0$ soils (Table 9).

## DISCUSSION

The lowest cumulative $N_2O$ emission was recorded in the biochar treated soils and at a lesser magnitude in the straw treated soils, whereas the highest $N_2O$ emission was observed in the no carbon treated soils. In both cases, the highest rate of N fertilizer recorded the greatest $N_2O$ emission. In contrast, *Chatskikh (2007)* and *Kammann et al. (2012)* reported that $N_2O$ fluxes were significantly increased by addition of biochar, particularly when added with mineral N-fertilizer. It has been shown that the type and rate of fertilizer have an important impact on $N_2O$ emissions (*Bouwman, Boumans & Batjes, 2002*). Some studies have reported that use of crop straw combined with mineral nitrogen fertilizer enhances soil quality while reducing $N_2O$ emissions (*Xu, Han & Ru, 2019*; *Sainju, 2016*). Crop straw return commonly aims at improving soil carbon and nitrogen cycling (*Xu, Han & Ru, 2019*; *Meng et al., 2017*), thought it can also be a source of trace gas emissions (*Cha*

**Table 8  Biomass yield of spring wheat as affected by different treatment.**

| Treatment | | Biomass yield (kg ha$^{-1}$) | | | |
|---|---|---|---|---|---|
| C source | N rates | 2014 | 2015 | 2016 | Mean |
| No carbon | $N_0$ | 2776d | 3030d | 2455d | 2754c |
| | $N_{50}$ | 3102c | 3358bcd | 3022c | 3161bc |
| | $N_{100}$ | 3399bc | 3739b | 3267bc | 3468b |
| Biochar | $N_0$ | 3295bc | 3530bc | 3147bc | 3324b |
| | $N_{50}$ | 3489b | 3767b | 3331bc | 3529b |
| | $N_{100}$ | 4291a | 4630a | 3788a | 4236a |
| Straw | $N_0$ | 3170bc | 3312cd | 3118bc | 3200bc |
| | $N_{50}$ | 3403bc | 3765b | 3365b | 3511b |
| | $N_{100}$ | 4082a | 4345a | 3633a | 4020b |

**Notes.**
Values with different letters within a column are significantly different at $P < 0.05$.

**Table 9  Grain yield of spring wheat as affected by different treatments.**

| Treatment | | Grain yield (kg ha$^{-1}$) | | | |
|---|---|---|---|---|---|
| C source | N rates | 2014 | 2015 | 2016 | Mean |
| No carbon | $N_0$ | 1305d | 1500d | 1009d | 1271d |
| | $N_{50}$ | 1538cd | 1896bc | 1043cd | 1492bcd |
| | $N_{100}$ | 1770abc | 1927bc | 1144cd | 1614cd |
| Biochar | $N_0$ | 1603bcd | 1789cd | 1124cd | 1505bcd |
| | $N_{50}$ | 1905abc | 2133b | 1233bc | 1757abc |
| | $N_{100}$ | 2139a | 2456a | 1567a | 2054a |
| Straw | $N_0$ | 1502cd | 1658cd | 1111cd | 1424cd |
| | $N_{50}$ | 1852abc | 1944bc | 1182cd | 1659bc |
| | $N_{100}$ | 1975ab | 2180ab | 1380ab | 1845ab |

**Notes.**
Values with different letters within a column are significantly different at $P < 0.05$.

*et al., 2016*). Nitrogen fertilization has the greatest potential to increase $N_2O$ emissions because mineral N controls both nitrification and denitrification. Other studies (*Zhang et al., 2011*) have shown that biochar combined with N-fertilizer can significantly reduce $N_2O$ emissions. One mechanism that may explain lower (cumulative) $N_2O$ fluxes from biochar + N-fertilizer-amended soils is the fact that relatively low C soils treated with N-fertilizer and biochar may retain relatively higher amounts of mineral N than soils untreated with N-fertilizer (*Zhang et al., 2011*). Nitrogen thereby retained provides a source of available N for plant uptake, which reduces N availability for microbes involved in denitrification processes. Since biochar has significant impact on soil environment and affects many soil parameters such as the availability of substrates (*Van Zwieten et al., 2009*), it is very likely that biochar will have significant effects on the production of $N_2O$. Their results is confirmed by the increased plant N uptake in this study (Table S4). *Singh et al. (2010)* reported that biochar can also reduce the N availability to microorganisms by absorption. In this study, improved soil porosity could also explain the decreased $N_2O$ emission recorded

when biochar was applied with N fertilizer. Soil aeration and improved porosity inhibit denitrification. Nitrogen dynamics are affected by changes in soil aeration, pH and the C/N ratio of the material incorporated into the soil. Biochar may suppress $N_2O$ production from denitrification by increasing the air content of the soil or by absorbing water from the soil, thus improving aeration of the soil (*Yanai, Toyota & Okazaki, 2007*). *Karhu et al. (2011)* shared similar view and observed that biochar amendment modifies soil physical properties such as reducing soil bulk density or increasing water holding capacity (*Karhu et al., 2011*), thereby increasing soil aeration.  This may lead to lower soil $N_2O$ emissions, as soil aeration influences both nitrifier and denitrifier activity. Soils, which are not affected by compaction often exhibit adequate porosity and therefore the risk of denitrification is lower compared with soils that have impaired infiltration or internal drainage (*Antille, 2018*). In this study, lower $N_2O$ emissions were also observed on the straw treated plots, although the effects were lesser relative to the biochar treated soils. The lower $N_2O$ emission under straw treated soils could be attributed to the accumulation of organic matter on the soil surface that led to reduced bulk density and thus improved soil aeration.

Reductions in $CH_4$ emission were observed in biochar–amended soils and to a lesser extent on straw amended soils compared to their controls. Literature evidence indicated that biochar input to soil can potentially reduce $CH_4$ emissions (*Yeboah et al., 2018*). In contrast, *Xie et al. (2021)* showed that charcoal input into soil may increase soil methane fluxes. The mechanisms underlying changes in soil $CH_4$ emissions following biochar amendment are unclear (*Lehmann et al., 2011*). The greater uptake of $CH_4$ may be attributed to the protected environment created for the $CH_4$ oxidizers and improved soil porosity. In this study, the greater uptake of methane in the soils with carbon amendment, particularly biochar amended soils with N fertilizer may be attributed to the favorable environment created for the $CH_4$ oxidizers. The aerobic, well drained soils can be a sink for $CH_4$ due to the possible high rate of $CH_4$ diffusion and ensuing oxidation by methanotrophs. Combined application of biochar and inorganic N-fertilizer in this study improved soil physical properties (reduction in soil bulk density and increased soil saturated hydraulic conductivity) . Such improved soil structural conditions are known to protect the ecological niche for methanotrophic bacteria, influence the gaseous diffusivity, and affect the rate of supply of atmospheric $CH_4$ (*Hütsch, 1998*; *Serrano-Silva et al., 2014*; *Ma et al., 2016*). Aerobic, well–drained soils behave as a sink for $CH_4$ due to the high rates of $CH_4$ diffusion and subsequent oxidation by methanotrophs (*Serrano-Silva et al., 2014*). However, these results do not appear to support the conclusions of *Laird (2008)* on the reduction observed in methane emissions from field plots, which was deduced as an increased $CH_4$ oxidation activity. Other studies have reported significant increase in $CH_4$ emissions following biochar or biomass application (*e.g.*, *Wang et al., 2012*). The authors explained that, the increased availability of labile C substrates following biochar or biomass addition stimulates the activities of methanogenic bacteria which may account for increased $CH_4$ emissions. However, this could be a short-term effect since labile carbon fraction in the materials could be mineralized rapidly (*Wang et al., 2012*).

The results of this study indicate that when biochar was applied together with fertilizer N, both biomass and grain yield of spring wheat increased. This finding shows the potential

of biochar applied together with fertilizer N to improve nutrient use efficiency in spring wheat in semiarid environment (*Solaiman et al., 2010*). Diverse reasons have been given to the positive effect of biochar applied in combination with fertilizer N on crop yield. *Bruun et al. (2011)* reported that combined application of biochar and N fertilizer has the potential to improve soil properties and could therefore be responsible for the effect observed. Similarly, both *Borchard et al. (2012)* and *Tammeorg et al. (2014)* attribute increased crop productivity when biochar is applied together with N fertilizer to improve nutrient availability. In the current study, increased yield may be attributed to increased nutrient availability and improved soil physical and chemical properties (soil bulk density, saturated hydraulic conductivity and soil organic carbon), as reported in earlier work (*Zhang et al., 2010*). These results imply that, when biochar and inorganic fertilizers are applied together, an increased nutrient supply to plants may be the most important factor in increasing crop yields. The higher biomass and grain yield obtained on the carbon amended soils compared to the soils without carbon in this study is attributed to the fact that in drier soils,crop residues provide a better soil environment by reducing temperature, conserving water, and improving soil quality resulting in better yield (*Zou et al., 2016*). Positive effects of biochar combined with N fertilizer on increasing SOC and hydraulic conductivity as well as decreasing soil bulk density was observed in this study. Therefore, this study evidenced a positive effect of biochar amendment on soil quality and spring wheat yield consistent over three consecutive years. Furthermore, the lowest yield recorded on the no carbon soils throughout this study may be related to the removal of all the aboveground biomass at the end of the cropping season. *Zhang, Yang & Wu (2008)* showed that field practices with low carbon inputs to arable soils as crop biomass removal and manure abandonment deplete soil organic carbon and reduce crop productivity. Therefore, when biochar was applied and crop residues retained, it had immediate effect and the beneficial influence on biomass and grain yields were obtained.

## CONCLUSIONS

Application of crop residue amendments combined with nitrogen fertilizer has been increasingly recommended as an effective management practice for mitigating greenhouse gas emissions while enhancing soil fertility, thereby increasing crop production. In this paper, we have shown that application of carbon amendment, especially biochar combined with N fertilizer in wheat grown under rain fed conditions in a semi-arid environment reduced nitrous oxide and methane emissions whilst increasing biomass and grain yield. This study confirmed our hypothesis that increased C inputs would increase the soils ability to reduce $N_2O$ and $CH_4$ soil emissions whiles increasing biomass and grain yield. The main conclusions derived from this work are: application of biochar + N-fertilizer ($BN_{100}$) or straw + N-fertilizer ($SN_{100}$) increased saturated hydraulic conductivity to significantly greater extent than the other treatments tested. This translated into higher biomass production and therefore grain yield in those treatments. These results indicate the dependency of crop yield on soil quality and imply that crop productivity could be increased without resource degradation when biochar is applied combined with N-fertilizer.

Application of biochar + N-fertilizer showed relatively lower $N_2O$ emissions, including increased uptake of $CH_4$, but the effect of $BN_{100}$ was consistently greater. The findings of this study suggest that biochar applied together with N-fertilizer can concurrently improve soil physical and chemical properties as well as biomass and grain yield while reducing the effect of agricultural activities on the environment. Based on this results, the potential exist for developing crop and soil management interventions around biochar applied together with fertilizer N in semiarid environments. Further studies that focus on $N_2O$ and $CH_4$ measurements after every rainfall, tillage and fertilization events are required for better recommendations.

## ACKNOWLEDGEMENTS

Our thanks go to Gansu Agricultural University for the assistance in the laboratory analysis of greenhouse gases and students and supervisors of Gansu Provincial Laboratory for assistance in laboratory analyses.

### Funding

This research was financially supported by the Natural Science Foundation of Gansu province (20JR10RA543), the National Natural Science Foundation of China (41661049,31571594), and the Scientific Research Start-up Funds for Openly-Recruited Doctors (GAU-KYQD-2018-39). The funders had no role in study design, data collection and analysis, decision to publish, or preparation of the manuscript.

### Grant Disclosures

The following grant information was disclosed by the authors:
The Natural Science Foundation of Gansu province: 20JR10RA543.
The National Natural Science Foundation of China: 41661049, 31571594.
The Scientific Research Start-up Funds for Openly-Recruited Doctors: GAU-KYQD-2018-39.

### Competing Interests

The authors declare there are no competing interests.

### Author Contributions

- Stephen Yeboah conceived and designed the experiments, performed the experiments, analyzed the data, prepared figures and/or tables, authored or reviewed drafts of the paper, and approved the final draft.
- Wu Jun performed the experiments, analyzed the data, prepared figures and/or tables, and approved the final draft.
- Cai Liqun conceived and designed the experiments, authored or reviewed drafts of the paper, and approved the final draft.
- Patricia Oteng-Darko and Erasmus Narteh Tetteh analyzed the data, prepared figures and/or tables, authored or reviewed drafts of the paper, and approved the final draft.

- Zhang Renzhi conceived and designed the experiments, performed the experiments, analyzed the data, authored or reviewed drafts of the paper, and approved the final draft.

## Data Availability

The raw data for $N_2O$ and $CH_4$, grain and biomass yield, soil data and Plant N are available in the Supplemental Files.

## Supplemental Information

Supplemental information for this article can be found online at http://dx.doi.org/10.7717/peerj.11937#supplemental-information.

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
