# Peer review of "Nitrous oxide, methane emissions and grain yield in rainfed wheat grown under nitrogen enriched biochar and straw in a semiarid environment"

_PeerJ, doi:10.7717/peerj.11937_

## Round 0.1 · original submission · Major Revisions

Our advisors have commented on your manuscript and they have suggested that you revise your manuscript before further consideration for publication in PeerJ. While assessing the comments from reviewers and reading your manuscript thoroughly, I feel that the following changes also need to be considered while preparing the revised manuscript:

(1) As clear from the experimental design, carbon (C) and nitrogen (N) are main factors, and the lack of data on soil organic C and soil mineral N contents is hurting this otherwise very interesting study.

(2) I also urge the authors to rethink their statistical analysis.

(3) I also don’t see any strong rationale of measuring only saturated hydraulic conductivity without presenting data on other soil physico-chemical properties.

(4) Discussion section also needs further strengthening.

(5) Resolution of the Figures must be improved for clarity.

(6) Check the manuscript extensively for grammatical and typographic mistakes.

Reviewer 1 ·

Basic reporting

The authors use clear language throughout that is generally easy to understand. However, there are a few typos (e.g. "whiles" - ln. 87) that could benefit from a more careful proofread. Further some phrases are a bit confusing, or could be streamlined to improved reader comprehension, like the objectives and the first sentence of the discussion.

My biggest issue with this manuscript is the explanations of the potential mechanisms for biochar and biomass to reduce GHG emissions provided in the introduction and the discussion of these mechanisms in the discussions. Further, I think that the scientific literature is not particularly well utilized to support the hypothesis or the results. I believe at least one more paragraph is needed in the introduction to more thoroughly explain how biochar/ biomass might reduce GHG emissions. The discussion needs much more support and a check to ensure the references are correctly cited in relation to their overall findings and how they relate to the study at hand.

The figures were a bit hard to interpret with 8 different patterns. Perhaps a more hierarchical color scheme where similar levels of C and N share features to make them more distinguishable on different levels?

Experimental design

The study is well designed and data well reported, however, the knowledge gap being filled by this data is less clear. In lines 80-84 the authors say "Much effort has been spend..." and "The literature is replete...", then introduce their knowledge gap with what reads as a vague restating what was listed as well known in the literature. I do believe the authors have a point as to the utility of data regarding biochar compared with straw, but it is unclear what they mean by "without the confounding effects of crops...(ln. 85)". This would greatly benefit from more exposition of the information informing their central hypothesis.

Validity of the findings

Comprehension of these findings would be aided significantly by a reorganization of their presentation in the tables and figures.

·

Basic reporting

This paper entitled “Nitrous, methane emissions and grain yield in rainfed wheat grown under nitrogen enriched biochar and biomass in a semiarid environment” is submitted for publication as an original research article in PeerJ journal. The study aims to examine the effects of biochar and straw amendments coupled or not to different levels of N fertilizer on soil emissions of N2O and CH4 as well as on wheat crop productivity. The novelty of this work is that it contributes to a better understanding of organic amendments impact (biochar in particular) on GHG emissions by cultivated soils under arid conditions, which is not frequently studied. Authors concluded to a reduction in nitrous oxide and methane emissions (with a larger extent for biochar+ highest N dose), associated to higher wheat yields. These findings can contribute to guiding agronomic practices related to soil fertilization and mitigation of climate change.
The manuscript is well written in a clear and organized way. Results were well described and correctly discussed.The references used by authors are relevant and support the discussed ideas. Conclusions are well stated and reply to the objectives announced by the authors.

Experimental design

Comments are given in "General comments for authors"

Validity of the findings

Comments are given in "General comments for authors"

Additional comments

L 68 : « A number of mechanisms have also been proposed in literature to explain the effect of biochar amendment on soil N2O and CH4 emissions, with limited amounts of evidence to support them. » Authors are requested to give more details about these mechanisms and provide references.

L 75 : Authors explained that the effects of biochar vary depending on biochar type and soil conditions and concluded that « Such responses have limited widespread use of biochar in cropping lands. » I think this affirmation should not be announced this way because there is a growing number of studies investigating the impact of biochar on soils and crops. Besides, this practice is becoming more and more encouraged in cultivated lands in many regions of the world.

It was mentioned in the background section, but it is also worth emphasizing in the introduction on the knowledge gap about biochar effects in arid environments. This is an additional source of novelty of this study, which avoids to the study to be of local interest. Authors can cite the following reference (https://link.springer.com/article/10.1007/s12517-018-4166-2) which shows that little is known about biochar application under arid conditions.


General comment : N2O and CH4 emissions as affected by biochar amendment were monitored during this study. What about CO2 ?


Study site :
L 106 : please specify the aridity index used here.

Experimental design
L 125 : « The two C sources (biochar and straw) were applied at the same quantity of the material based on the straw returned to the soil every year. ». In my understanding, so that we would compare biochar and straw treatments, the amounts of biochar and straw applied to soil should bring the same quantity of carbon. But when I performed calculations, I found that biochar brings to soil 15t x %C = 6 tons of C while straw brings 4.5t x 3 years X %C = 8 tons. Is this difference because you have taken into consideration straw C mineralization ? Please further explain that in the text.

L130 : please specify the nature of feedstocks used for biochar production and pyrolysis holding time if applicable. What’s the size of biochar granules applied to soil ?


Soil sampling and analyses
Were soil characteristics (other than Ksat) determined after straw or biochar amendment ? Bulk density (indicator of soil porosity) and pH for example are important parameters to be monitored, because they affect soil conditions, which affect in turn soil microorganisms development and activity. Why was Ksat the only soil property monitored during this study ?

L56 : Omit the e after CO2
L 244-245 treatments with s ans were instead of was.
L252 : the results were clear
L339 : reduce

Discussion
L 298 : Provide references supporting this first mechanism.

Were N crop content monitored ? This can help checking the hypothesis of higher uptake of nitogen by crops which leads to less N available for denitrification.

---

## Round 0.2 · Minor Revisions

Your revised manuscript has been assessed by our reviewers who agree that manuscript is improved but still requires some important changes. I invite you to revise the manuscript by giving considerations to reviewer comments.

·

Basic reporting

Most of my comments were addressed by authors

Experimental design

Most of my comments were addressed by authors

Validity of the findings

Most of my comments were addressed by authors

Additional comments

Most of my comments were addressed by authors

·

Basic reporting

The study is interesting that how to reduce the use of fertilizer and keep environment safe from GHG emission with out compromising crop yield, but the article need significant improvement as GHG emission in semi arid region of china has been explored by many researcher, therefore i suggest to the author that please improve your introduction and discussion section by adding recent and relevant references. Please revised your article to avoid English grammar mistakes.

Experimental design

Correct

Validity of the findings

The presentation of the results are good but the conclusion is poorly written.

Additional comments

Line 40-41: Please rephrase to avoid grammatical mistakes.
Line 60-61: Please add a reference
Please add recent references in the introduction section
Soil sampling, measurements and analyses: Please write in detail

Discussion: In semi-arid region and Loess Plateau, many researchers have done alot of work on GHG emission (organic farming). I suggest to re-write this section with proper and recent citations and please avoid English grammar mistakes. The current form of discussion is not acceptable,

Conclusion: It looks like a summary, please come to the point, Please write about your main significant results and your recommendations, and missing gap for future.

















;

---

## Round 0.3 · accepted · Accept

I am pleased to inform you that I have accepted your manuscript for publication.